# Links of Cytoskeletal Integrity with Disease and Aging

**DOI:** 10.3390/cells11182896

**Published:** 2022-09-16

**Authors:** Yu Jin Kim, Min Jeong Cho, Won Dong Yu, Myung Joo Kim, Sally Yunsun Kim, Jae Ho Lee

**Affiliations:** 1CHA Fertility Center Seoul Station, Jung-gu, Seoul 04637, Korea; 2Department of Biomedical Sciences, College of Life Science, CHA University, Pochen 11160, Korea; 3National Heart and Lung Institute, Imperial College London, London SW7 2AZ, UK

**Keywords:** aging, mechanical property, cytoskeleton, disorder, rejuvenation

## Abstract

Aging is a complex feature and involves loss of multiple functions and nonreversible phenotypes. However, several studies suggest it is possible to protect against aging and promote rejuvenation. Aging is associated with many factors, such as telomere shortening, DNA damage, mitochondrial dysfunction, and loss of homeostasis. The integrity of the cytoskeleton is associated with several cellular functions, such as migration, proliferation, degeneration, and mitochondrial bioenergy production, and chronic disorders, including neuronal degeneration and premature aging. Cytoskeletal integrity is closely related with several functional activities of cells, such as aging, proliferation, degeneration, and mitochondrial bioenergy production. Therefore, regulation of cytoskeletal integrity may be useful to elicit antiaging effects and to treat degenerative diseases, such as dementia. The actin cytoskeleton is dynamic because its assembly and disassembly change depending on the cellular status. Aged cells exhibit loss of cytoskeletal stability and decline in functional activities linked to longevity. Several studies reported that improvement of cytoskeletal stability can recover functional activities. In particular, microtubule stabilizers can be used to treat dementia. Furthermore, studies of the quality of aged oocytes and embryos revealed a relationship between cytoskeletal integrity and mitochondrial activity. This review summarizes the links of cytoskeletal properties with aging and degenerative diseases and how cytoskeletal integrity can be modulated to elicit antiaging and therapeutic effects.

## 1. Introduction

The elderly population has dramatically increased worldwide. A United Nations report states that the percentage of the world population older than 85 years will triple within 30 years [1]. Most advanced nations have severe public health issues for aging societies where they cannot live alone [2]. Elderly people with health problems may be helped by family members or government support [2]. This leads to an economic burden for all generations to maintain the welfare of the entire population. Therefore, much scientific and medical research focuses on aging and maintaining the health of older people. Aging increases the risk of degenerative disorders, such as dementia, autoimmune diseases, and vision loss [3].

Research of aging focuses on two concepts: (1) antiaging to overcome aging-related diseases and (2) rejuvenation to reverse aging. Rejuvenation is more complex and riskier for clinical applications [4,5]. It is important to investigate the cause of aging for both concepts. The cause remains unclear because aging involves multiple factors and types of processing. Furthermore, the phenotype of aging is extremely heterogeneous according to race, nation, location, and each individual person [6].

Aging has several phenotypes, such as genomic instability, telomere attrition, loss of proteostasis, deregulated nutrient sensing, mitochondrial dysfunction, cellular senescence, stem cell exhaustion, and altered intercellular communication [7]. In molecular biology, increased genomic mutation and alteration of telomere length are the main factors involved in aging [8]. Telomere length decreases during aging. However, maintenance of telomere length due to upregulation of telomerase causes tumorigenesis [8,9]. Therefore, molecular control of telomere length to elicit antiaging effects is very complex and involves multiple factors. Further studies are required to elucidate the mechanism underlying control of telomere length. Other aging phenotypes, such as loss of proteostasis, deregulated nutrient sensing, and stem cell exhaustion, are associated with mitochondrial dysfunction [7]. Dysfunctional mitochondria are related to an elevated level of reactive oxygen species (ROS), which cause mutation of the mitochondrial genome and protein damage, and thereby perturb metabolic regulation [10]. Most aging phenotypes are related with mitochondrial dysfunction. Therefore, scientists interested in antiaging effects and aging-related diseases are attempting to recover mitochondrial function. Chemical-based materials, such as anti-ROS agents (e.g., NAD), have been developed to overcome mitochondrial dysfunction. Several anti-ROS products have been applied to elicit antiaging effects. However, a study reported that anti-ROS agents cannot overcome aging of human oocytes [11]. Other potential factors must be studied and further evidence must be obtained regarding the role of mitochondrial dysfunction in aging. 

A recent study reported that dynamic mitochondrial behavior, including the balance between fusion, fission, and movement, may play pivotal roles in regulation of mitochondrial activity [12]. Mitochondria are transported along the actin cytoskeleton by specific motor proteins. The mechanical properties of the cytoskeleton significantly differ between old and young cells [13]. The cytoskeleton exhibits increased stiffness and a decreased capacity to reversibly form in old cells. The mechanical properties of the cytoskeleton are important for transfer of mechanical signals [14]. Therefore, both mechanical properties and cell stiffness linked with the cytoskeleton may be associated with mitochondrial functional activity in aging cells [15]. However, the role of the actin cytoskeleton in mitochondrial dysfunction related with aging is unclear [16]. 

There are several lines of evidence that a relationship exists between the actin cytoskeleton and aging [17]. Perturbed integrity of the actin cytoskeleton is connected with loss of mitochondrial function and aging. Several scientists have attempted to elucidate how modulation of the cytoskeleton can control aging [15]. Chemical and biological stimuli and mechanical signals from extracellular environments can regulate cellular behaviors. Biophysical signals modulate the mechanical properties of the cytoskeleton [18]. The mechanical properties of cells regulate cellular behaviors, such as proliferation, differentiation, and apoptosis. The mechanical properties of the cytoskeleton dramatically change with aging [19,20,21]. Artificial disturbance of the cytoskeleton induces aging phenotypes, such as slow proliferation and increased mitochondrial dysfunction. Aged cells treated with a cytoskeleton stabilizer exhibit reversal of aging phenotypes due to recovery of mitochondrial functional activities [19]. Taken together, these findings demonstrate that the cytoskeletal stability is a key factor for reversal of aging. Here, we review the relevance of the cytoskeleton to aging at the cellular level in various organs and current methodologies to promote healthy aging. 

## 2. Description of the Cytoskeleton

### 2.1. Actin Cytoskeleton

Actin is the most abundant protein and a key cytoskeletal component in most eukaryotic cells. It is highly conserved and participates in more protein-protein interactions than any other known protein. These properties, along with its ability to transition between monomeric (G-actin) and filamentous (F-actin) states under the control of nucleotide hydrolysis, ions, and many actin-binding proteins, make actin a critical player in many cellular functions, ranging from cell motility and maintenance of cell shape and polarity to regulation of transcription. As an important part of the cytoskeleton, actin is essential for cell stability and morphogenesis. Dynamic regulation of the F-actin cytoskeleton is critical for numerous physical cellular processes, such as cell division, endocytosis, and cell migration [22]. These processes are regulated by the dynamic mechanical environment via cross-linking of microtubules and microfilaments at membranes and in the nuclear zone. 

### 2.2. Microfilaments

Microfilaments are also called actin filaments because they are mostly composed of actin; their structure comprises two strands of actin wound in a spiral. They are ~7 nm thick, making them the thinnest cytoskeletal filaments. Microfilaments have many functions [23]. They aid cytokinesis, which is division of the cytoplasm when a cell divides into two daughter cells, and cell motility, and allow single-celled organisms, such as amoebas, to move. Microfilaments are also involved in cytoplasmic streaming, which is the flow of cytosol (the liquid part of cytoplasm) throughout the cell. Cytoplasmic streaming transports nutrients and organelles. Microfilaments are also part of muscle cells and allow these cells to contract, along with myosin. Actin and myosin are the two main components of muscle contractile elements [24].

### 2.3. Microtubules

Microtubules are the largest cytoskeletal fibers, forming ~23 nm wide hollow tubes composed of polymerized α-, β-, and γ-tubulin dimers. Microtubules form structures such as flagella, which are “tails” that propel a cell forward. They are also found in structures such as cilia, which are appendages that increase a cell’s surface area and, in some cases, allow the cell to move. The γ-microtubules in animal cells emanate from an organelle called the centrosome, which is a microtubule-organizing center. The centrosome is located near the middle of the cell, and microtubules radiate out from it. Microtubules are important for forming the spindle apparatus (or mitotic spindle), which separates sister chromatids so that one copy is transferred to each daughter cell during cell division. 

Microtubules are highly dynamic polar filaments composed of α/β-tubulin dimers that undergo active polymerization and depolymerization, which are essential for cell survival and function [25]. Each microtubule is composed of protofilaments with a long cylindrical shape, and interactions between protofilaments result in stiffness and resistance to bending forces [26]. Microtubules have an essential role in cell structure and motility and, therefore, microtubule networks are dynamic and continuously remodeled [24]. The rapid growth and shortening of the end of an individual microtubule is referred to as dynamic instability [26,27]. This process is regulated by several factors, including GTP hydrolysis [26]. GTP hydrolysis induces a bend in the subunit and destabilizes the microtubule, enabling the protofilament to adopt a curled conformation [23]. Persistent growth of microtubules is interrupted by occasional rapid shrinkage, which is referred to as catastrophe, while switching from shrinkage to growth is referred to as rescue [24]. Dynamic instability is modified when a rescue factor copolymerizes with tubulin or binds to the positive ends of microtubules [28].

### 2.4. Intermediate Filaments

Intermediate filaments are ~8–12 nm wide; they are called intermediate because they are larger than microfilaments but smaller than microtubules. They are composed of different proteins, such as keratin (found in hair, nails, and animals with scales, horns, or hooves), vimentin, desmin, and lamins. All intermediate filament proteins are found in the cytoplasm except for lamins, which are found in the nucleus and help support the nuclear envelope. Intermediate filaments in the cytoplasm maintain cell shape, bear tension, and provide structural support to the cell.

Intermediate filaments are distributed throughout the cytoplasm and inner nuclear membrane [29]. In contrast with actin and microtubules, which contain globular proteins, intermediate filaments comprise fibrous proteins [29]. Intermediate filaments lack polarity but are flexible and elastic, and have high tensile strength, making them the most stable cytoskeletal component [29]. Strong interactions between protofilaments enable intermediate filaments to resist compression and bending forces, and thus they can absorb mechanical stress without breaking [29]. 

### 2.5. The Strutural Formation of Actin-Cytoskeleton and Mitochondrial Transfer 

Actin-cytoskeleton has dynamic assemble and dissemble formulation through nucleation, elongation, and polymerization depending on the cell status [30]. In particular, actin formation is regulated by nucleation-promoting factors (NPFs), such as formin and the Arp2/3 complex. [31] (Figure 1). Moreover, actin formation is also controlled by extracellular environment sensing, such as cell adhesion receptor, tyrosine kinases, and actin modulator [32]. These key protein signals link with actin cytoskeleton rearrangement and adhesion dynamics to several cell physiological processes, such as migration, proliferation, differentiation, and apoptosis [33]. In addition, actin mediates the movement of a cellular organ, such as the mitochondrion [34]. Myosin motors couple to mitochondria to link with actin by an unknown mechanism [35]. 

Therefore, microtubules provide a major back-bone structure in cells and actin-cytoskeletons support cellular mechanical signal transfer to nuclei. These are very important properties of cells for regulation of cellular physiology, such as cell viability or death. Moreover, microtubules are composed of positive-end proteins for retrograde transport from nuclei and negative-end protein to nuclear-side polymerization [26]. In addition, the role of microtubule polymerization is an important element for the migration, proliferation, and differentiation of cells [36]. Therefore, microtubules provide a line to mitochondria transfer in the cytoplasm of cells. Mitochondrial motility is required for asymmetric distribution of the organelle during cell behavior, enrichment of the organelle at sites of high ATP utilization, and mitochondrial inheritance [37]. Mitochondria movement is associated with several different motor proteins and the cytoskeleton. Microtubule acts as a major transfer line as it cross-links mitochondria with dynactin/dynein and kinectin/kinesin (Figure 1). Both kinesin and dynein are motors for mitochondria and mediate bidirectional transport by either controlling motor activity and number or engaging each motor with microtubules [38] (Figure 1). Mitochondrial dynamics and transport have emerged as key factors in the regulation of neuronal differentiation and survival [39]. Mitochondria are dynamically transported in and out of axons and dendrites to maintain neuronal and synaptic function [40]. The integrity of microtubule is regulated by phosphorylation of tau protein [41]. In addition, the loss of integrity of microtubule is associated with aging and degeneration of cells [42]. 

Intermediate filaments, together with actin and microtubules, support mechanotransduction signal and regulation of several cell behaviors [43]. However, intermediate filaments are not involved in mitochondria transfer [44].

## 3. Cytoskeletal Integrity Changes with Disease and Aging

The cytoskeleton determines the structure and morphology of cells and contributes to their mechanical properties [45]. This section provides an overview of the role of each cytoskeletal component and how it is altered by aging.

### 3.1. Altered Expression of Actin during Aging

Actin filaments influence the shape, stiffness, and movement of the cells [29]. Cellular movement is controlled by the strength and type of interactions with neighboring cells or the extracellular matrix (ECM) [46]. Actin filaments interact with myosin motor proteins to enable important processes, such as cytokinesis and muscle contraction [47]. The actin cytoskeleton generates intracellular forces required for many cellular functions, including cell motility, cell division, cytokinesis, muscle contraction, and intracellular transport [48]. Mutations of actin or actin-binding proteins or treatment with actin-binding drugs can alter the dynamic state of actin, which leads to changes in cell fate [16]. Increased turnover of F-actin leads to cell longevity, whereas decreased actin turnover may lead to cell death through an apoptosis-like pathway [16,49]. Actin turnover is decreased in aged cells [16].

Aging disrupts the organization and dynamics of the actin cytoskeleton by altering expression of actin, leading to aging-related diseases [15]. The causes of altered actin expression include alteration of hormonal, nutritional, and actin cytoskeleton polymerization activity statuses [15]. The actin cytoskeleton has important roles in cell division and apoptosis; therefore, dysfunction of actin due to aging leads to uncontrolled cell proliferation and tumorigenesis [15]. Actin dysfunction and altered integrity of the actin cytoskeleton also disrupt proper functioning of reproductive cells, which explains the challenges of in vitro fertilization with aged eggs [50]. There is robust evidence that the actin cytoskeleton determines the mechanical properties of the cell surface [51], which is important to consider in aging studies because one of the hallmarks of aging cells is altered mechanical properties. Treatment with jasplakinolide, which polymerizes actin filaments, results in stiffer cells, demonstrating that actin polymerization increases cell stiffness [51]. Treatment with blebbistatin, a specific inhibitor of myosin II activity, reorganizes the actin cytoskeleton and alters the mechanical properties of cells [51].

### 3.2. Damage of Actin by Reactive Oxygen Species (ROS) and Aging

Actin cytoskeletal signaling networks comprise numerous proteins, such as integrins, small GTPases, kinases, phosphatases, ion channels, and transporters [52]. These proteins directly regulate actin assembly, organization, and function, and also indirectly influence cell growth and survival by impacting signaling cascades and networks [52]. Physiological concentrations of ROS are required for normal cellular functions, including thyroid hormone synthesis, calcium homeostasis, ion channel dynamics, and cytoskeletal remodeling [53]. An imbalance of chemical reduction and oxidation can lead to dysregulation of actin and microtubule dynamics. Studies in yeast found that changes in actin turnover and accumulation of large F-actin aggregates and microtubule instability can increase ROS levels in the cytosol [16,54]. ROS are produced by mitochondria, as evidenced by the finding that accumulation of ROS in cells is reduced, along with decreased actin dynamics, when mitochondrial function is disrupted [16]. When certain amino acid residues in microtubules and actin microfilaments are oxidized, the ability of microtubules to polymerize is reduced and actin microfilaments are severed [53]. By contrast, inhibition of ROS production can cause aberrant actin polymerization and cellular functions. 

In the case of neurons, abnormalities in critical processes, such as outgrowth of neurites and development and polarization of neurons, can occur when ROS production is inhibited [53]. Production of ROS increases with age; however, it is unclear whether ROS-induced damage is the cause of aging [55]. Increased oxidation in neurons is associated with aging, oxidative stress, and diseases such as Alzheimer’s disease (AD), Parkinson’s disease, Huntington’s disease, and amyotrophic lateral sclerosis [53]. Oxidative stress affects actin cytoskeleton reorganization by directly modifying the actin cytoskeleton or actin regulatory proteins, resulting in alteration of cytoskeletal dynamics and the ability of G-actin to polymerize [56,57]. ROS can alter actin cytoskeleton signaling by oxidizing actin regulatory proteins or modulating protein or mRNA expression [52,56]. For example, ROS can alter expression of integrin αv and α5 subunits in endothelial cells, which, in turn, affects cell adhesion to the ECM [58].

In immune cells, physiological levels of ROS regulate actin dynamics required for chemotaxis and cell migration, while an increased level of ROS negatively regulates actin polymerization [59,60]. An increased level of ROS drives glutathionylation of a critical cysteine residue (Cys374), which reduces the ability of G-actin to polymerize into F-actin [13,59]. By contrast, ROS depletion decreases actin glutathionylation, which increases the amount of F-actin and ultimately impairs cell migration [59]. In mice deficient of glutaredoxin-1, an enzyme that catalyzes actin deglutathionylation, significantly fewer neutrophils are recruited to sites of inflammation [59]. In addition, the same study showed that the ability of neutrophils to kill bacteria is reduced in glutaredoxin-1-deficient mice, leading to the conclusion that ROS-induced actin glutathionylation and its modulation by glutaredoxin-1 are key processes controlling actin dynamics in neutrophils that ultimately influence innate immunity and host defense [59]. Other studies showed that silencing of glutaredoxin-1 induces cell senescence [61] and glutaredoxin-1 activity decreases with aging [62,63]. 

Idiopathic pulmonary fibrosis (IPF) is a fatal lung disease associated with aging because it is most commonly diagnosed when people are older than 60 years and its incidence increases exponentially with each decade of life [62,64,65,66]. The pathophysiologic mechanisms of IPF are similar to the mechanisms underlying normal aging-related changes [65]. The hallmarks of IPF include accumulation of scar tissue in the lungs, which is characterized by excessive production of type I collagen-rich matrix by lung myofibroblasts, resulting in loss of lung function [66]. The progressive stiffening of lung tissue in IPF is attributed to the increased stiffness of the cytoskeleton in fibroblasts [67]. IPF fibroblasts are stiffer than healthy control fibroblasts and, upon treatment with transforming growth factor-β1, a profibrotic cytokine that induces cytoskeletal rearrangement in fibroblasts, the stiffening response of IPF fibroblasts is almost 15 times higher than that of healthy control fibroblasts [67]. Moreover, morphological analysis revealed that IPF fibroblasts have a lower actin density, diffuser distribution of actin, and weaker colocalization of actin with the cell membrane than healthy control fibroblasts [66]. Lungs experience mechanical insults as part of normal breathing, and proper organization of the actin cytoskeleton in pulmonary cells is critical for their integrity and functionality [66]. Therefore, decreased levels of actin and altered cytoskeletal regulation in IPF fibroblasts are key contributing factors in the course of this disease and may be promising targets for novel treatments.

### 3.3. Alterations in Microtubule Regulation with Aging

Microtubule density decreases with aging [42], and abnormal regulation of microtubules is linked with aging-related disorders [68]. Microtubule regulation is critical in neurons because their polarized morphologies are primarily maintained by properly functioning microtubules [68]. Precise regulation of microtubule and actin dynamics is required for numerous functions of neurons and maintenance of neuronal structure, which can deteriorate with age and lead to neurodegenerative disorders, such as Parkinson’s disease and AD [15,68]. In Parkinson’s disease, the assembly and stability of microtubules are compromised. In AD, the number and total length of microtubules in neurons are significantly reduced due to malfunction of tau, a major microtubule-associated protein in neurons that stabilizes microtubules [42]. Hirano bodies are intracellular protein aggregates composed of F-actin and actin-associated proteins that are found in the brains of patients with neurodegenerative diseases and in aged individuals [69]. They can modulate cell death in the presence of tau protein [69]. The interaction between tau and actin can lead to tau-induced neurodegeneration [69]. The complexity of the relationship between tau and actin is evidenced by the fact that minor phosphorylation of tau may enhance actin binding, while phosphorylation of tau at certain sites reduces its ability to associate with actin filaments and microtubules [39,70,71]. Phosphorylated tau has important functions, including microtubule stabilization, actin reorganization, and synaptic activity [51]. Phosphorylation of tau is a key event that connects tau with amyloid-β (Aβ) protein during progression of AD; therefore, controlling specific phosphorylation events may be a potential therapeutic strategy [68].

Detyrosination of microtubules regulates mechanotransduction by altering the mechanical properties of the cytoskeleton [72]. Microtubule detyrosination is a post-translational modification that occurs after polymerization and involves cleavage of a C-terminal tyrosine residue by a tubulin carboxypeptidase [72]. This process stabilizes microtubule filaments and increases their association with other cytoskeletal elements, which, in turn, increases cytoskeletal stiffness. Treatment with pharmacological agents, such as nocodazole, which prevents microtubule polymerization and destabilizes microtubules, can alter cytoskeletal stiffness, resulting in softer cells compared with control untreated cells [51,73].

During tissue development, internal genetic and external biochemical cues orchestrate the developing cell structure and cellular mechanical properties that ultimately impact cellular behavior and organ function [45,51]. In a study that compared changes in mechanical properties of outer hair and pillar cells isolated from the murine inner ear at embryonic and postnatal time points, stiffness of these cells increased with increasing passage number [51]. In parallel, changes of the actin cytoskeleton and microtubules were observed using fluorescence microscopy together with phalloidin, which labels the actin cytoskeleton, and an antibody raised against acetylated tubulin to label a subset of stable microtubules because there is a high correlation between microtubule stability and acetylation [51]. The relative fluorescence intensity of phalloidin staining was 72% higher in postnatal (P5) outer hair cells than in embryonic (E16) outer hair cells but the staining intensity of acetylated tubulin was similar, whereas the staining intensities of phalloidin and acetylated tubulin were 69% and 80% higher in P5 pillar cells than in E16 pillar cells, respectively [51]. Therefore, it is evident that cells become stiffer due to developmental changes in the cytoskeletal network and signaling cascades (e.g., fibroblast growth factor signaling that mediates cell structural development) that alter cytoskeletal dynamics.

Changes in mechanical properties of human skin are a key characteristic of the aging process [74]. In addition to changes in collagen and elastin organization and ECM density, dermal fibroblasts from aged donors are 60% stiffer than those from younger donors [74]. The increased stiffness of older fibroblasts is attributed to increased actin polymerization, leading to an increased level of actin filaments [74]. Moreover, that study showed there is a possible correlation between increased cell stiffness and a higher microtubule cytoskeleton content, although this was statistically insignificant. These data were obtained by quantifying the fluorescence intensity of microtubules using fluorescence-activated cell sorting (FACS) after fixation and permeabilization of suspended cells. Future studies using more samples or live cells and conjugated antibodies for FACS may lead to statistical significance.

The microtubule network impacts the viscoelasticity of human myocardium [59]. Impaired left ventricular relaxation is a hallmark of heart failure because an increased density of microtubules increases stiffness in the failing myocardium [59]. Therefore, a therapeutic approach that reduces myocardial stiffness by targeting microtubules in cardiomyocytes could potentially prevent heart failure, a disease that dramatically increases with age [59,60,61]. Viscoelasticity during diastolic stretch is higher in cardiomyocytes isolated from failing hearts than in healthy control cardiomyocytes [75]. Targeted pharmacological treatment using colchicine, a microtubule-inhibiting drug, reduces the microtubule density by inducing depolymerization of cardiac microtubules and improves myocardial performance in animal models [76,77,78].

The relationship between microtubule and mitochondria dynamics has been previously reported [79] (Figure 2). Their regulation is under the control of several factors, such as mTOR, and AMPK, that alters microtubule integrity [80,81]. In particular, mTOR is a regulator of cell growth and metabolism and it interacts with mitochondria through the translation of nucleus-encoded mitochondrial mRNAs, which increase mitochondrial ATP production [81]. It is also involved in mitochondrial fission depending on the environment. mTOR modulates the cytoskeleton, including microtubule integrity for cell proliferation and migration. mTOR is associated with microtubule stability for mitochondria movement [82]. Moreover, microtubule stabilizer is derived from the optimal integrity of microtubules and enhances spare respiration capacity (SRC) of mitochondria in the cells. A high SRC level represents healthy ATP productivity of mitochondria.

In Figure 3, microtubule integrity dramatically differs between young and old oocytes [65]. Aged oocytes mature slower than young oocytes. Aging is a complex process governed by several factors [17], and mitochondria are crucial components of the cellular aging process [83]. Our results indicate that mitochondrial motility, maturation, and aging are associated in oocytes. Young oocytes exhibited higher dynamic mitochondrial motility during germinal vesicle breakdown than aged oocytes [79]. The location of mitochondria substantially changed according to the maturation stage and group, with higher concentrations of mitochondria in nuclear areas at the metaphase I stage in young oocytes. By contrast, aged oocytes exhibited low mitochondrial motility and did not display preferential accumulation of mitochondria in nuclear areas during maturation. These observations are consistent with the finding that the location of mitochondria changes depending on the stage of the aging process [79]. In particular, the cytochalasin-B-treated young oocytes reduced the mitochondrial movements, which resembled the characteristics of the aged oocytes (Figure 2). Therefore, actin cytoskeleton instability is a primary factor in age-related variations in the mitochondrial activity in oocytes. Our data revealed that immobility of mitochondria reduced the amount of biological energy required to support oocyte maturation. This phenomenon may be associated with the stability of the actin cytoskeleton during maturation, resulting in aneuploidy in embryos. In general, dysfunctional mitochondria contribute to the aging process. 

### 3.4. Changes in Intermediate Filaments with Aging

An important function of intermediate filaments is to regulate signaling that controls cell survival and growth [84]. Intermediate filaments also contribute to regulation of tissue repair and regeneration upon injury [84]. Alterations in the composition and organization of the intermediate filament network due to tissue injury change cellular viscoelastic properties, which enables optimal migration of regenerating cells into injured sites [84]. Intermediate filament proteins in focal adhesions also mediate integrin-based mechano-transduction in response to the stiffness of the ECM [85]. Vascular aging increases the stiffness of the ECM and vascular smooth muscle cells (VSMCs) [86]. Vimentin is an intermediate filament protein and is highly expressed in endothelial cells and VSMCs in larger arteries, such as the aorta and carotid artery [85]. It is involved in regulating the functions of endothelial cells and VSMCs and contributes to aging-related arterial stiffness [68]. Furthermore, mRNA and protein levels of vimentin are increased in senescent cells, indicating that vimentin has a role in aging [87]. Aging impairs expression of Jagged1, a Notch signaling ligand that regulates the vascular wall structure, in endothelial cells after vascular injury and enhances VSMC proliferation [88]. Notch signaling involves actomyosin contractility, which requires cytoskeletal forces that contribute to signaling between adjacent cells [88]. Modifications of vimentin regulate the strength of Notch signaling and structural remodeling of arteries through an interaction with Jagged1 in response to hemodynamic stress [88].

In human skin, vimentin is a specific target of the glycation reaction, which is a chemical reaction between an amino acid and a reducing sugar [29,89]. Glycation of vimentin filaments induces formation of perinuclear aggregates of glycated vimentin protein, which leads to loss of the contractile capacity of skin fibroblasts [29,89]. This is more commonly observed in fibroblasts isolated from elderly patients who have higher levels of vimentin glycation, leading to features of aged skin, including loss of contractile functions [29,89]. 

Disruption of the structures of actin and intermediate filaments alters the mechanical properties of chondrocytes by decreasing the elastic modulus and viscoelastic properties [90]. In chondrocytes, intermediate and actin microfilaments are linked through plectin, which is involved in co-ordinating different cytoskeletal elements and contributes to the mechanical properties of individual cells [90]. Mild disruption of intermediate filaments can lead to loss of this interaction, resulting in alteration of the actin cytoskeleton structure [90]. Aging leads to deterioration of articular cartilage due to wear and tear, trauma, or disease, which can result in the development of osteoarthritis [90]. The stiffness of chondrocytes increases with age [90,91]. This, together with aging-associated changes in the ECM, influences the synthetic activity of chondrocytes [90].

Beaded filament structural proteins 1 and 2 are intermediate filament proteins in the eye lens that have a role in lens physiology and disease [92]. In particular, both proteins have roles in aging processes in the lens because their mutations cause cataracts, which are highly prevalent in the aged population [92]. Changes in lens proteins occur with aging, while loss of antioxidants and the capacity to scavenge free radicals exposes lens proteins to the risk of oxidation [93]. Furthermore, changes in the elastic properties of the lens due to alteration of lens proteins contribute to increased lens stiffness with aging, which plays a role in the development of cataracts [93]. During cataract formation, cytoskeletal components in the lens, including beaded filament proteins, are targeted for proteolysis, and loss of beaded filaments causes cataracts [92,94]. Downregulation of α-crystallin, which is required to stabilize the structures of the actin cytoskeleton, microtubules, and intermediate filaments, is observed in the epithelium of aged lens and can promote cataract formation [95].

## 4. Diseases Related to Cytoskeletal Integrity and Cytoskeleton-Targeting Therapy 

### 4.1. Dementia

Dementia is a major illness among elderly people. AD is the most common form of dementia. Pathologically, AD is characterized by amyloid plaques and neurofibrillary tangles in the brain, with associated loss of synapsis and neurons, resulting in cognitive deficits and eventually dementia [96,97]. Aβ peptide and tau protein are the primary components of plaques and tangles, respectively [96]. In the decades since Aβ and tau were identified, the development of therapies for AD has primarily focused on Aβ, but tau has received more attention in recent years, partly due to the failure of various Aβ-targeting treatments in clinical trials. Alteration of cytoskeletal integrity is linked with the development of dementia. Several research groups have developed cytoskeleton-targeting (tau-targeting) therapies for AD [41]. Initially, potential anti-tau therapies were mainly based on inhibition of kinases or tau aggregation or on stabilization of microtubules, but most of these have been discontinued because of toxicity and/or lack of efficacy. Currently, most tau-targeting therapies in clinical trials are immunotherapies, which have shown promise in numerous preclinical studies. Given that tau pathology correlates better with cognitive impairments than Aβ lesions, targeting of tau is expected to be more effective than clearance of Aβ once clinical symptoms are evident [97]. With future improvements in diagnostics, these two hallmarks of AD might be targeted prophylactically.

### 4.2. Diminished Reproductive Potential in Oocytes with Aging 

Maternal age affects the actin cytoskeleton and is associated with several novel criteria for the quality of oocytes [50]. Mitochondrial functional activity in oocytes is important for normal chromosomal segregation and assessment of oocyte quality and embryo viability. Older women have lower pregnancy rates and a higher chance of producing an embryo with a chromosomal abnormality than younger women. Aneuploidy ratios significantly increase with aged gametes and zygotes. In general, aging of zygotes cannot be reversed. Dysfunctional mitochondria are a key element in aged zygotes. We found an association between cytoskeletal instability and reduced mitochondrial function, which is related to mitochondrial dynamics, in oocytes of mice of advanced age [79]. These data suggest that microtubule stability is involved in regulation of mitochondrial properties, including migration and proliferation, contrary to commonly held views regarding mitochondrial bioenergy production in mainstream biology. A previous report clearly demonstrates that microtubule stability enhances bioenergetic activity of mitochondria [79]. Therefore, microtubule stabilization in aged oocytes and embryos may be a possible therapeutic strategy to overcome infertility in older women.

### 4.3. COPD

COPD is a representative chronic inflammatory disease. It is characterized by progressive and irreversible airflow obstruction, which may be caused by long-term exposure to cigarette smoke or inflammation-stimulating exogenous pathogens. The prevalence of COPD, a representative chronic inflammatory disease, increases in elderly populations, and COPD is considered to be a condition of accelerated lung aging [98].

Among the several phenotypes of COPD, remodeled or compromised cytoskeletal organization is a key feature in addition to mitochondrial dysfunction. D’Anna et al. demonstrated that airflow was limited in a human bronchial epithelial cell line (16HBE) exposed to cigarette smoke extract and lipopolysaccharide (LPS) due to excessive F-actin polymerization and expression of tubulin-specific chaperone A (TBCA) and microtubule-associated protein RP/EB family member 1 (MARE1) [99]. They demonstrated that co-exposure to cigarette smoke and LPS aberrantly increases coactosin-like protein (COTL-1) expression, which reduces cytoskeletal stability. Heijink et al. reported decreased mRNA expression of junctional proteins in airway epithelium, which increases mucosal permeability [100]. They found that loss of epithelial barrier function is mainly due to the disrupted localization of occludin, claudin, and ZO-1, which establish tight junctions in epithelium. Moreover, Bidan et al. showed that airway smooth muscle cells are stiffened by reinforcement of their actin cytoskeleton in response to changes in the ECM and chronic inflammatory conditions [101]. Contracted airway layers due to alteration of cytoskeletal organization in epithelial and smooth muscle cells of airway structures limit fluent air exchange and cause severe rupture of the airways in the lungs. Although many researchers have tried to establish the disrupted organization in the microenvironment of COPD, more comprehensive studies are required. For example, the dynamic interaction between cytoskeletal organization and other cellular phenotypes, such as mitochondrial activity and gene expression patterns, should be investigated further. Our recent study demonstrated that cytoskeletal organization and mitochondrial activity were altered in human placenta-derived mesenchymal stem cells (hPD-MSCs) that had been exposed to the proinflammatory cytokines, tumor necrosis factor-α (TNF-α) and interferon-γ (IFN-γ) [102]. Interestingly, an anti-inflammatory chemical, called MIT-001 or NecroX-7, simultaneously improved both mitochondrial homeostasis and cytoskeletal organization in hPD-MSCs exposed to TNF-α and IFN-γ. The alignment of epithelial cells and cilia on their surface is another crucial feature of the airway structure that is influenced by cytoskeletal regulation. The three-dimensional structure of the airway epithelium has a critical role in air exchange and pathogen exclusion [103]. Therefore, future studies that more exquisitely and comprehensively investigate cytoskeletal organization may propose a novel therapeutic strategy for COPD [104]. 

## 5. Conclusions

The cytoskeleton not only influences formation of the cellular structure, but also has several critical roles in cell proliferation, degeneration, inflammation, and mitochondrial signaling. Various recent studies reported that the cytoskeleton has several important cellular functions and is linked with mitochondrial functional activity and neuronal degeneration. Moreover, optimal cytoskeletal integrity is linked with life span, longevity, and aging combined with the functional mitochondria activity. In this review, we summarized the functional activity of the cytoskeleton and development of cytoskeleton-targeting therapies. Taken together, we tried to figure the understanding from reviewing current knowledge regarding integrity of cytoskeleton’s relationship with disease and aging (Figure 2). There is enormous potential to develop new medicines that target the functional activity of the cytoskeleton for diseases that currently lack treatments of degeneration and antiaging. 

## Figures and Tables

**Figure 1 cells-11-02896-f001:**
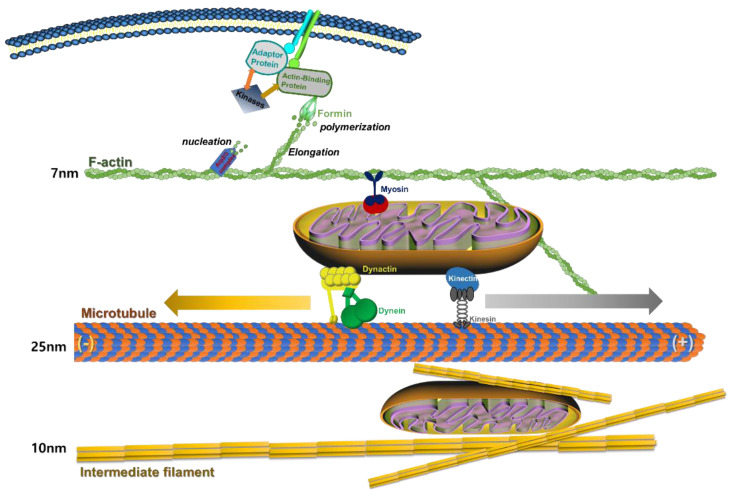
The structural property and modulation of actin-cytoskeleton and mitochondria transfer.

**Figure 2 cells-11-02896-f002:**
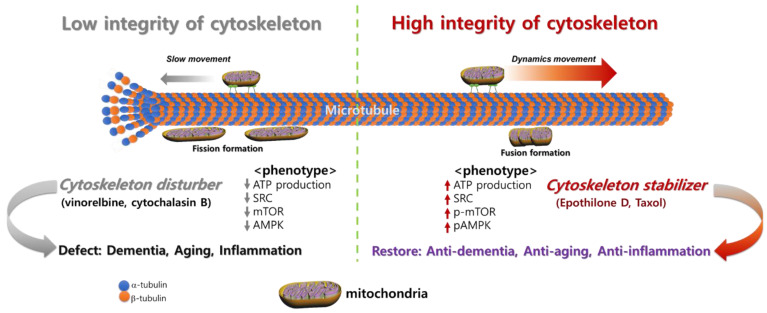
Degeneration, aging, and inflammation associated with integrity of microtubules and dynamic movement of mitochondria.

**Figure 3 cells-11-02896-f003:**
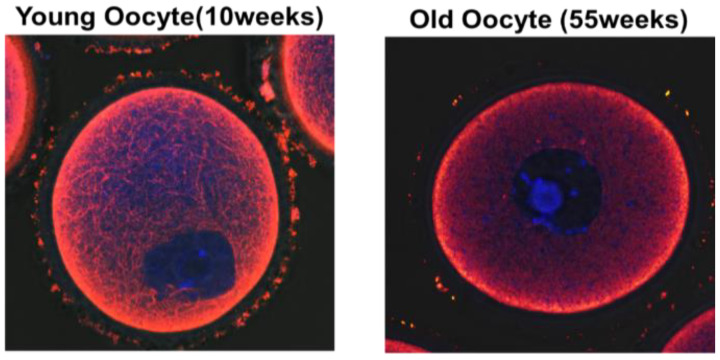
Immunocytochemistry fluorescence staining with anti-microtubule antibody (red) and nuclear count staining with Hoechst 33342 (blue) in the young (10 weeks) and old mouse oocyte (55 weeks).

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
