# Peer review of "Links of Cytoskeletal Integrity with Disease and Aging"

_cells, 2022, doi:10.3390/cells11182896_

Round 1

Reviewer 1 Report

Aging is a complex process, which is associated with many factors. In this review, Kim and colleagues describe the connection between aging, diseases and the cytoskeletal integrity. In their overview, the authors summarize components othe cytoskeleton and change of cytoskeletal integrity is related to aging and multiple diseases. I recommend that the following points are considered in improving the manuscript prior to publication:

1.    Change of cytoskeletal integrity is related to the aging process, and it also affects the life span. Thus, it is better to add a paragraph to discuss about it, for example: link of life span and the cytoskeletal integrity.

2.     For the title of 3.1, I suggest replacing it with “Altered expression of actin during aging”.

Author Response

1. Change of cytoskeletal integrity is related to the aging process, and it also affects the life span. Thus, it is better to add a paragraph to discuss about it, for example: link of life span and the cytoskeletal integrity.
R) Thank you for your nice comment, we added some sentences in the discussion following your comment.

2. For the title of 3.1, I suggest replacing it with “Altered expression of actin during aging”.
R) Thank you for giving us a great comment. I will update the text. 

Reviewer 2 Report

It is an interesting topic for writing a review but the current version is pre-mature for publication to "Cells". The review does not go much in to the details and it does not provide or illustrate any detailed cellular mechanism on the regulation of cytoskeleton affecting aging. Important structural components of MT and actin cytoskeleton are not shown or mentioned. The actin regulatory proteins with key roles in the apoptotic process due to aging are not mentioned either.  However, the authors made an abundant point in the section related to diseases and cytoskeletal integrity.

Hereafter, some specific comments and suggestions for the authors to consider.

1. Line 14. It is unclear the " various targeting models ". I would suggest the authors to mention the models or to omit this context.

2. Lines 15- 19. "The integrity of the cytoskeleton is associated with cellular functions and chronic disorders such as neuronal de generation and premature aging. Cytoskeletal integrity is closely related with several functional activities of cells such as aging, proliferation, degeneration, and mitochondrial bioenergy production. " The text is redundant and confusing. I would suggest to read as: 

The integrity of the cytoskeleton is associated with cellular functions such as proliferation, degeneration, and mitochondrial bioenergy production and chronic disorders including neuronal degeneration and premature aging.

3. Line 117. The authors refer to microtubules (MT) and explain their structure by refering to α/β-tubulin. They are missing important components of MT such as the γ-tubulin, capping proteins, MT binding proteins etc. For example γ-Tubulin is essential for the centrosome formation mentioned in line 121.

4. Line 117-118. "  Microtubules are the largest cytoskeletal fibers at ~23 nm thick. They are hollow tubes composed of α- and β-tubulin". ....  I would suggest :

Microtubules are the largest cytoskeletal fibers forming ~23 nm wide hollow tubes composed of polymerized α- and β-tubulin dimers.

5. Line 167-169. "Increased turnover of F-actin leads to cell longevity. It is a very strong argument. It has to be revised. The cited paper is a review from 2005. It refers mainly to several eukaryotic cells that implicates the actin cytoskeleton as a regulator of ROS release from mitochondria to activate cell death pathways. These yeast studies proposed just a link of actin to apoptosis and ageing. The authors should include more recent literature on this aspect.

6. The "description of the cytoskeleton" is simple and does not provide information of key components. Important structural components of MT and actin cytoskeleton are not shown or mentioned. There are elegant in-vivo studies from invertebrate (like drosophila) development or from macrophages, for example, showing how different actin networks (linear or branched) are activated and regulated by PTPs/RTK signaling to induce vital cellular functions like nutrient endocytosis, metabolism etc. Formins and the activators of the Arp2/3 complex, termed nucleation-promoting factors (NPFs) required for actin assembly in cells are missing. The actin regulatory proteins with key roles in the apoptotic process due to aging are not mentioned either. How mitochondria travel in the MT are not shown. I would strongly recommend to illustrate MT and F-actin assembly (& de-assembly) and their basic components in a separate figure (figure 2) together with a panel showing the MT / mitochondria association and how are transported into the MT. This it would be easier for the reader to understand and follow the Scheme 1.

7. Line 465. The authors provide a summary of current knowledge regarding integrity of cytoskeleton and its relationship with disease and aging. The Scheme 1 illustrates how MT integrity (and not the F-actin networks) can be associated with the dynamic movement of mitochondria to affect disease. The authors should expand their conclusions and interpret how F-actin networks can affect the related disease and aging. Otherwise they have to change the title of the review.

8. Actin assembly is promoted in lung cancer cells, upon depletion of PTPN3 for example, leading to the enhancement of cancer cell migration/invasion and metastasis. Considering then that actin assembly enhances other disease like cancer, the authors have to address in the conclusion how the development of drugs promoting actin assembly to treat aging related disease will not affect cell migration and cancer.

9. The Scheme 1 is not explained in the legends or in the text with details. The α- and β-tubulin dimers should be indicated. The phenotypes of low or high integrity of MTs (-mTOR, SRC, etc) are not explained or mentioned in the text.

10. The Figure 1 is not indicated in the text. The authors have to mention in the legends the anti-microtubule antibody used and the species of the oocytes presented.

Author Response

Hereafter, some specific comments and suggestions for the authors to consider.
1. Line 14. It is unclear the " various targeting models ". I would suggest the authors to mention the models or to omit this context.
R) Thank you for your comment. We will delete this sentence. 

2. Lines 15- 19. "The integrity of the cytoskeleton is associated with cellular functions and chronic disorders such as neuronal de generation and premature aging. Cytoskeletal integrity is closely related with several functional activities of cells such as aging, proliferation, degeneration, and mitochondrial bioenergy production. " The text is redundant and confusing. I would suggest to read as: The integrity of the cytoskeleton is associated with cellular functions such as proliferation, degeneration, and mitochondrial bioenergy production and chronic disorders including neuronal degeneration and premature aging.
R) Thank you for your kind comment. We have replaced our sentence according to your suggestion.

3. Line 117. The authors refer to microtubules (MT) and explain their structure by refering to α/β-tubulin. They are missing important components of MT such as the γ-tubulin, capping proteins, MT binding proteins etc. For example γ-Tubulin is essential for the centrosome formation mentioned in line 121.
R) Thank you for your comment. We have put in the γ-tubulin information on the text.

4. Line 117-118. " Microtubules are the largest cytoskeletal fibers at ~23 nm thick. They are hollow tubes composed of α- and β-tubulin". ....  I would suggest : Microtubules are the largest cytoskeletal fibers forming ~23 nm wide hollow tubes composed of polymerized α- and β-tubulin dimers.
R) Thank you for your comment. We have replaced that sentence according to your comment.

5. Line 167-169. "Increased turnover of F-actin leads to cell longevity. It is a very strong argument. It has to be revised. The cited paper is a review from 2005. It refers mainly to several eukaryotic cells that implicates the actin cytoskeleton as a regulator of ROS release from mitochondria to activate cell death pathways. These yeast studies proposed just a link of actin to apoptosis and ageing. The authors should include more recent literature on this aspect.
R) Thank you for your comment. We have updated the reference for that sentence.

6. The "description of the cytoskeleton" is simple and does not provide information of key components. Important structural components of MT and actin cytoskeleton are not shown or mentioned. There are elegant in-vivo studies from invertebrate (like drosophila) development or from macrophages, for example, showing how different actin networks (linear or branched) are activated and regulated by PTPs/RTK signaling to induce vital cellular functions like nutrient endocytosis, metabolism etc. Formins and the activators of the Arp2/3 complex, termed nucleation-promoting factors (NPFs) required for actin assembly in cells are missing. The actin regulatory proteins with key roles in the apoptotic process due to aging are not mentioned either. How mitochondria travel in the MT are not shown. I would strongly recommend to illustrate MT and F-actin assembly (& de-assembly) and their basic components in a separate figure (figure 2) together with a panel showing the MT / mitochondria association and how are transported into the MT. This it would be easier for the reader to understand and follow the Scheme 1.
R) Thank you for your valuable comment. We updated the new illustration regarding the structure and modulation of actin-cytoskeleton assembly/de-assembly and the mitochondria transfer following your comment. Thanks again. 

7. Line 465. The authors provide a summary of current knowledge regarding integrity of cytoskeleton and its relationship with disease and aging. The Scheme 1 illustrates how MT integrity (and not the F-actin networks) can be associated with the dynamic movement of mitochondria to affect disease. The authors should expand their conclusions and interpret how F-actin networks can affect the related disease and aging. Otherwise they have to change the title of the review.
R) We updated the movement of the mitochondria to affect diseases like neurodegeneration disease in section 2.5 and in the conclusion. 

8. Actin assembly is promoted in lung cancer cells, upon depletion of PTPN3 for example, leading to the enhancement of cancer cell migration/invasion and metastasis. Considering then that actin assembly enhances other disease like cancer, the authors have to address in the conclusion how the development of drugs promoting actin assembly to treat aging related disease will not affect cell migration and cancer.
R) Thank you for your comment. Based on our study and some references, the integrity of actin-cytoskeleton related aging oocyte and microtubule stabilizer treated cells enhanced functional activity of mitochondria compare with disturber treated cells. We have reported that the microtubule instability is one of the most important primary factors in the mitochondrial activity of cells. Mitochondrial motility and loss of function activity may be related to microtubule instability in the HEK293(Cells 2021, 10, 3600. https://doi.org/10.3390/cells10123600). You have suggested that PTPN2 leads actin assembly enhancement of cancer cell migration for metastasis. PTPN3 is related with the actin assembly and disassembly process for cell migration of cancer cells. It is an important role for the actin. But we want to review about optimal stability in the manuscript and focus on the integrity of cytoskeleton related with degeneration, aging and inflammation. 

9. The Scheme 1 is not explained in the legends or in the text with details. The α- and β-tubulin dimers should be indicated. The phenotypes of low or high integrity of MTs (-mTOR, SRC, etc) are not explained or mentioned in the text.
R) Thank you for your comment. We updated and explained the dimer of MT and mTOR, SRC as regulation factor and means of SRC in the text. 

10. The Figure 1 is not indicated in the text. The authors have to mention in the legends the anti-microtubule antibody used and the species of the oocytes presented.
R) We updated and provided an explanation regarding figure 1 in the text according to your comments. 

Round 2

Reviewer 2 Report

The authors have addressed the concerns raised in the initial view and no further revision is needed.

As minor comments: 1) It would be easier for naive readers in biology if the authors indicate the microtubules and mitochondria in the panel (Scheme 2).

2) I recommend that the authors do a critical screening of the manuscript for grammatical errors. For example, in lines 157 and 195 should be stated "structural formation" and not "structure formation" ; And "structural property" and not " structure property".

Author Response

The authors have addressed the concerns raised in the initial view and no further revision is needed.
As minor comments: 
1) It would be easier for naive readers in biology if the authors indicate the microtubules and mitochondria in the panel (Scheme 2).
R) We updated your comment in the scheme 2

2) I recommend that the authors do a critical screening of the manuscript for grammatical errors. For example, in lines 157 and 195 should be stated "structural formation" and not "structure formation" ; And "structural property" and not " structure property".
R) We updated all follow your comment